

# A clustering effectiveness measurement model based on merging similar clusters

Guiqin Duan[1,2] and Chensong Zou[3]

[1] School of Computer and Information Engineering, Guangdong Songshan Vocational and Technical College, Shaoguan, China
[2] Shaoguan Ecological and Cultural Big Data Engineering & Research Center, Shaoguan, China
[3] Shaoguan Automation Engineering Research and Development Center, Shaoguan, China

## ABSTRACT

This article presents a clustering effectiveness measurement model based on merging similar clusters to address the problems experienced by the affinity propagation (AP) algorithm in the clustering process, such as excessive local clustering, low accuracy, and invalid clustering evaluation results that occur due to the lack of variety in some internal evaluation indices when the proportion of clusters is very high. First, depending upon the "rough clustering" process of the AP clustering algorithm, similar clusters are merged according to the relationship between the similarity between any two clusters and the average inter-cluster similarity in the entire sample set to decrease the maximum number of clusters $K_{max}$. Then, a new scheme is proposed to calculate intra-cluster compactness, inter-cluster relative density, and inter-cluster overlap coefficient. On the basis of this new method, several internal evaluation indices based on intra-cluster cohesion and inter-cluster dispersion are designed. Results of experiments show that the proposed model can perform clustering and classification correctly and provide accurate ranges for clustering using public UCI and NSL-KDD datasets, and it is significantly superior to the three improved clustering algorithms compared with it in terms of intrusion detection indices such as detection rate and false positive rate (FPR).

## INTRODUCTION

The cluster analysis technique is widely used to perform data analysis, and it is an unsupervised learning method. Its main function is to perform the division of a set of unlabeled data into multiple clusters so that data points in the same cluster are as similar to each other as possible and those in other clusters are as different from each other as possible (*Jain, Murty & Flynn, 1999*). The classical K-means clustering algorithm (*MacQueen, 1967*), the K-medoids clustering algorithm (*Park & Jun, 2009*), and the fuzzy C-means clustering algorithm (*Dunnâ, 1974*) are common clustering techniques. Unlike these classical clustering algorithms, the affinity propagation (AP) algorithm (*Frey & Dueck, 2007*) does not need cluster center initialization, and it is fast and stable. It can outperform classical clustering algorithms, especially when large datasets need to be processed. Studies show that the AP clustering when performed on datasets with complex

Corresponding author
Chensong Zou,
zouchensong2003@126.com

structures, parameters $p$ greatly affect the clustering, and the resulting number of clusters is often higher than the actual value (*Wang, Chang & Du, 2021*; *Li et al., 2021*). Therefore, it is necessary to further analyze and evaluate the range and rationality of the number of clusters if the AP clustering algorithm is used. In addition, the evaluation of clustering quality is very important for cluster analysis. Commonly used evaluation methods determine the best clustering and classification results by comparing cluster validity indices (*Liang, Han & Yang, 2020*). The current study is focused on addressing the research question: What is the rationale behind utilizing the AP clustering algorithm to determine the optimal range of clusters, and how does this impact the quality of clustering compared to commonly used evaluation methods?

Researchers typically classify cluster validity indices (CVIs) into two groups, namely, internal indices and external indices. The main difference between these two groups of indices is whether external information is used. External CVIs require that standard class labels be before known because their main purpose is to perform the selection of the optimal clustering algorithm for a particular dataset. On the other hand, internal CVIs are usually used to select the optimum number of clusters for a given dataset without prior knowledge. In fact, prior knowledge about the class is not frequently available. Therefore, CVIs are the only option for clustering evaluation in such cases (*Guan & Loew, 2020*). For example, the optimal number of clusters may vary for different indices without the prior information. Which evaluation index has a K value with more reference value? This has become an important research topic in the field of cluster analysis. Classical internal CVIs such as DB, CH, XB, and IGP (*Huang et al., 2013*; *Guangli et al., 2022*) have been widely used. Studies have shown that these classical evaluation indices can effectively handle spherical clusters with simple structures, but it is difficult to achieve good results when they are used to process the data of non-spherical clusters. For this reason, researchers have successively proposed a series of evaluation indices, such as S_Dbw, CDbw, and DBCV (*Liang, Han & Yang, 2020*). To a certain extent, these evaluation indices solve the problem in the evaluation of clustering quality for non-spherical clusters. However, when the data becomes more complex, the cluster density varies or clusters overlap each other, some evaluation indices will be limited to a certain extent. Therefore, improving the existing evaluation indices or designing more scientific and reasonable evaluation indices has become a crucial research direction for the subject topic of clustering evaluation.

First, considering the deficiencies of the AP clustering algorithm and internal CVIs in their applications, the original AP clustering algorithm is improved by merging similar clusters to reduce the greater number of clusters $K_{max}$. Then, through a comparative study of the classical internal evaluation indices, a new method for calculating intra-cluster cohesion, inter-cluster overlap coefficient, and inter-cluster dispersion is given, and a new internal evaluation index is proposed. Subsequently, a model for evaluating clustering quality is designed on the basis of the improved AP clustering algorithm. Finally, the performance of this model is verified using the UCI data and NSL-KDD intrusion detection data. In summary, this study has the following contributions:

- This article proposes an enhanced AP clustering algorithm that reduces the number of resulting clusters by merging similar ones.
- This article proposes a novel internal evaluation index to calculate the intra-cluster cohesion, inter-cluster overlap coefficient, and inter-cluster dispersion.
- This article proposed to verify the performance of the improved AP clustering method using the detection rate, correct classification rate, and accuracy metrics such as FNR, and false positive rate (FPR) on the UCI and NSL-KDD datasets.

The remainder of the article is structured as follows:

The overview of the existing literature is given in 'Related Work'; the current status of the research regarding the AP clustering model is given in 'Materials & Methods Current Research Status'; while the proposed effective clustering measurement model is given in 'A New Clustering Effectiveness Measurement Model'. Next, comprehensive experiments with results and comparisons with other algorithms are presented in 'Experiments and Results'. Lastly, the proposed research is concluded with future research implications in 'Conclusion'.

## RELATED WORK

*Chandra, Canale & Dunson (2023)* proposed Bayesian clustering (Lamb) along with a set of low-dimensional latent variables. The proposed model is highly amenable to scaling the posterior inference and avoids the limitations of high dimensionality under mild assumptions. More complex data structures that do not involve real-valued vectors and allow for kernel misspecification can be handled by the proposed model's application.

The AP model, with its propagation capability, has been proposed to cluster low and high-influence indices (*Geng et al., 2019*). The application of the proposed model has been used to select and refine input indices. The model has great discriminant ability by screening the high-influence factors. However, due to excessive clustering, the AP model faces certain issues of low accuracy, excessive clustering, and invalid clustering results. Recent work *Duan & Zou (2023)* proposes a model by reducing the clustering and optimizing the maximum clustering number. The proposed model has been evaluated on the NSL-KDD dataset to demonstrate its strength for correct clustering partitioning.

The AP model has shown its applications in various research areas. AP clustering demonstrates high performance and low complexity for real-time and joint transmission in cloud radio access networks. The application of the AP model has shown normalized execution time and provides highly effective energies and spectral properties compared with existing clustering models (*Park et al., 2021*). The proposed AP model has the potential to be considered for more realistic joint transmissions, owing to the various arrival times.

Performing clustering on real-world large-scale data is a complex and time-consuming process. *Lin et al. (2019)* proposed a bottom-up clustering (BUC) technique to revolutionize convolutional neural networks. The proposed technique involved similarity within the same identity and diversity among different density factors. Identity over samples is used to group similar samples into one identity, and the data volume of each cluster is balanced by using bottom-up clustering. The results show that the proposed technique

is superior to semi-supervised learning and transfer learning techniques. Clustering models are of paramount importance due to their various applications. For example, the hierarchical clustering approach is used to categorize airports into a number of clusters (*Sheridan et al., 2020*). Detection of anomalous flights leverages the airports and safety measures. Event flights had an average anomalous score compared to non-event flights.

# MATERIALS & METHODS CURRENT RESEARCH STATUS

To facilitate the description of the AP clustering algorithm, various internal indices for clustering evaluation, and the algorithm proposed in this paper, the following sample set is created: $X = \{x_1, x_2, \ldots, x_i, \ldots, x_N\}$, where $N$ is the total number of samples, and the sample $x_i = \{x_{i1}, x_{i2}, \ldots, x_{il}\}$, where $l$ denotes the feature dimensionality of samples. $X$ is divided into $K$ clusters. Therefore, $X = \{C_1, C_2, \ldots, C_K\}$. The resulting set of cluster centers (centroids) is $V = \{v_1, v_2, \ldots, v_K\}$, $n_i$ is the number of samples in the $i^{\text{th}}$ cluster, and $c$ is the mean center of the sample set.

## AP algorithm

*Frey & Dueck (2007)* proposed the AP algorithm for clustering in SCIENCE in 2007. The main rationale behind the proposal of this clustering algorithm was to involve all data points as potential cluster centers and connect them to form a network. During its implementation, the cluster center of each data point is calculated through the transmission of information along each edge in the network (*Wang, Chang & Du, 2021*). The basic structure of the AP clustering algorithm is given as follows:

First, the similarity *s* between sample pairs is determined based on the Euclidean distance formula, and the similarity matrix $S$ for all samples is obtained. Next, the availability *a,* and responsibility *r* of the samples are iteratively updated. When the number of iterative updates exceeds the preset value or the updating of representative points ceases after multiple iterations, the AP clustering algorithm terminates. At this point, the remaining samples can be assigned to the appropriate clusters to complete the clustering process. The operation process of the AP clustering algorithm is detailed below.

Step 1: Use the opposite number of the Euclidean distance as the similarity $s$ $(i,k)$ between the samples $x_i$ and $x_k$ to obtain the similarity matrix $S$.

$$s(i,k) = \begin{cases} -\|x_i - x_k\| & i \neq k \\ p(k) & i = k \end{cases} \tag{1}$$

where $p$ $(k)$ is the value on the diagonal of $S$ that expresses the tendency of the sample $x_k$ to be selected as the cluster center. The larger the $p$ $(k)$ value, the greater the probability of the sample $x_k$ being selected as the representative of cluster centers. In this paper, reference is made to the practice recommended in *Halkidi & Vazirgiannis (2001)*, *i.e.,* taking the median of the similarity matrix $S$ as the default value of the bias parameter $p$ when no prior knowledge is available.

This article involves two key parameters bias parameter and damping parameter. A damping factor between 0 and 1 is applied to control the numerical oscillation. The data

points in a sample have the same possibility for becoming the clustering center and the parametric value of all data points is set to the same value of P. Certain parametric values of the noise level in data, convergence criteria, affinity matrix construction, and similarity matric could affect the proposed approach.

The parameter selection for the AP algorithm is based on a combination of domain knowledge and empirical experimentation. The outcome of the proposed approach can be affected by bias and damping parametric values. For example, a higher preference value encourages more data points to become exemplars and a lower bias value makes it more challenging and results in fewer and larger clusters. Similarly, a lower damping factor value makes faster convergence but can result in instability or oscillation in the cluster assignments.

Step 2: Transmit information, iteratively update the availability $a$, and responsibility $r$, and generate a representative cluster center. The responsibility $r(i,k)$ represents the degree of responsibility of $x_k$ to $x_i$. The larger the value of $r(i,k)$, the greater the probability of $x_k$ becoming the center of $x_i$'s cluster. The availability $a(i,k)$ represents the degree of availability of $x_i$ to $x_k$. The larger the value of $a(i,k)$, the higher the possibility of $x_i$'s cluster choosing $x_k$ as its center. The methods for iteratively updating the responsibility $r$ and availability $a$ are given in Eqs. (2) and (3).

$$r(i,k) = s(i,k) - \max_{k' \neq k}\{a(i,k') + s(i,k')\} \tag{2}$$

where $a(i,k')$ denotes the degree of availability of samples other than $x_k$ to $x_i$; $s(i,k')$ denotes the degree of responsibility of samples other than $x_k$ to $x_i$, i.e., the degree of completion of all samples other than $x_k$ for the ownership of $x_i$; $r(i,k)$ is the cumulative evidence proving that $x_k$ has become the center of $x_i$.

$$a(i,k) = \begin{cases} \min\left\{0, r(k,k) + \sum\limits_{i' \notin \{i,k\}} \max[0, r(i',k)]\right\} & i \neq k \\ a(k,k) = \sum\limits_{i' \neq k} \max[0, r(i',k)] & i = k \end{cases} \tag{3}$$

where $r(k,k)$ denotes the self-responsibility of $x_k$, $a(k,k)$ denotes the self-availability of $x_k$; $r(i',k)$ denotes the similarity level of $x_k$ serving as the centroids of all samples other than $x_i$; $a(i,k)$ represents the degree of possibility of taking all availability values greater than or equal to 0 plus the self-availability value of $x_k$ as the cluster center.

Oscillations can occur when the responsibility $r$ and availability $a$ are updated. The damping parameter $\lambda$ is introduced to reduce the amplitude of oscillation, eliminate oscillations, and correct $r(i,k)$ and $a(i,k)$ during iterations to make the iterative process more stable. The damping parameter is set to $\lambda \in [0, 1)$, and the number of iterations is $t$. The corrected iterative processes are expressed by Eqs. (4) and (5).

$$r(i,k)^{t+1} = (1-\lambda) \times r(i,k)^{t+1} + \lambda \times r(i,k)^t \tag{4}$$

$$a(i,k)^{t+1} = (1-\lambda) \times a(i,k)^{t+1} + \lambda \times a(i,k)^t \tag{5}$$

where $r(i,k)^t$ and $r(i,k)^{t+1}$ denote the degrees of responsibility for the $t^{th}$ and $t+1^{th}$ iterations, and $a(i,k)^t$ and $a(i,k)^{t+1}$ denote the degrees of availability for the $t^{th}$ and $t+1^{th}$ iterations.

Step 3: Determine the cluster-center representative. Select the sample $x_k$ with the largest sum of responsibility $r(i,k)$ and availability $a(i,k)$ as the representative point of the cluster to which $x_i$ belongs. The condition that the number of clusters $k$ should satisfy is given in Eq. (6).

$$k = \arg \max\{a(i,k) + r(i,k)\}. \tag{6}$$

From the operation process described above, it can be seen that the bias parameter $p$ appears when the responsibility $r(i,k)$ is calculated in Step 1. As the value of $p$ increases, $r(i,k)$ and $a(i,k)$ increase, and the probability of the candidate representative points becoming cluster centers also increases accordingly. It can be known that when the value of $p$ is large and there are a large number of candidate representative points, the tendency of more candidate representative points to be cluster centers will become increasingly obvious. However, the current theoretical basis for determining the value of $p$ is inadequate, which leads to the fact that the AP clustering algorithm usually takes the locally optimal solution or the approximate global optimum as its final result (*Zhou et al., 2021*). The practice of taking the median of similarity values as the value of $p$ has a certain reference value (*Li et al., 2017*). Still, the resulting value of K is often greater than the correct number of clusters, and the clustering accuracy of this method appears insufficient. Therefore, it is necessary to further analyze and evaluate the clustering quality using appropriate evaluation indices to achieve more ideal results. In addition, when the sample size is huge, the AP clustering algorithm will experience problems such as insufficient storage space and long running time, and when it is used to process non-cluster-shaped sample sets, it will usually produce a large number of local clusters. Depending on a locally linear embedding (LLE) hybrid kernel function, an innovative AP clustering method was proposed in *Sun et al. (2018)*, In addition to it, a new hybrid kernel was introduced to measure similarity and construct the similarity matrix for AP clustering. However, when the values of upper bounds for clustering obtained by this AP clustering method are large, this method will still face the problem of long running time. In a study *Gan, Xiuhong & Xiaohui (2015)*, the merge process was integrated into the AP clustering algorithm to merge the two clusters satisfying the condition that the minimum inter-cluster distance is less than the average distance between clusters in the entire dataset, thus improving the performance of the proposed algorithm and solving the problem of unsatisfactory clustering results with non-cluster-shaped datasets. However, because this algorithm uses the minimum inter-cluster distance, sample data with low similarity may be assigned to one cluster in the merge process.

## Internal evaluation indices

Internal evaluation indices have been widely used to perform the evaluation of the clustering quality for data without classification labels by measuring the intra-cluster and inter-cluster similarities after clustering using only the attributes of the dataset in question, and no

external information is needed in such an evaluation process. High-quality clustering can achieve high intra-cluster compactness and good inter-cluster separation. These indices are usually achieved through certain forms of combination based on the selection of values for intra-cluster and inter-cluster distances (by taking extreme values or by weighting after taking extreme values). Commonly used internal evaluation indices and their characteristics are analyzed below.

## Davies–Bouldin index

$$DB(K) = \frac{1}{K} \sum_{i=1}^{K} \max_{j,j \neq i} \frac{\frac{1}{n_i} \sum_{x \in C_i} d(x, v_i) + \frac{1}{n_j} \sum_{x \in C_j} d(x, v_j)}{d(v_i, v_j)} \tag{7}$$

where $K$ represents the number of clusters, $C_i$ denotes the $i$-th cluster, $n_i$ is the number of samples of the $i$-th cluster, $v_i$ refers to the cluster center of the $i$-th cluster, $d(x, v_i)$ is the Euclidean distance between the sample $x$ and the cluster center $v_i$, and $d(v_i, v_j)$ represents the Euclidean distance between the cluster centers $v_i$ and $v_j$.

The Davies–Bouldin (DB) index of a sample set is obtained by taking the sum of the average distances between the samples in two adjacent clusters and the centers of the two clusters as the intra-cluster distance (*Davies & Bouldin, 1979*). The smaller the value of the DB index, the lower the similarity between clusters and the better the clustering quality. This index is suitable for evaluating datasets that are characterized by high intra-cluster compactness and great inter-cluster distance, but when the degree of inter-cluster overlap in a dataset is high (for example, when data points are distributed in a circular pattern), it is very difficult to complete clustering evaluation accurately using this index.

## Xie-Beni index

$$XB(K) = \frac{\sum_{i=1}^{K} \sum_{j=1}^{N} u_{ij}^2 d^2(x_j, v_i)}{N \times \min_{i \neq j} d^2(v_i, v_j)} \tag{8}$$

where $N$ represents the total number of samples, and $u_{ij}$ refers to the fuzzy membership degree of sample $x_j$ and cluster $C_i$.

In the Xie-Beni (XB) index, the numerator represents the intra-cluster compactness for fuzzy partitioning, and the denominator represents inter-cluster separation (*Xie & Beni, 1991*). This index takes into account the geometric structure of the dataset to be processed, calculates the intra-cluster correlation, and considers the inter-cluster distance. To achieve the best clustering result, the difference in the same cluster must be very small. The greater the value of the numerator, the higher the degree of inter-cluster separation. However, when the number of clusters is excessively large, the numerator of the XB index gradually decreases with the increase in $K$. As the value of $K$ increases, the numerator of the XB index tends to 0, the value of the denominator increases continuously, and the XB index will lose the ability to judge due to its excessive monotony.

$V_{.P.C.}$ **and** $V_{.P.E.}$

$$V_{PC} = \frac{1}{N} \sum_{i=1}^{K} \sum_{j=1}^{N} u_{ij}^2 \qquad (9)$$

$$V_{PE} = -\frac{1}{N} \sum_{i=1}^{K} \sum_{j=1}^{N} u_{ij} \times \log u_{ij}. \qquad (10)$$

Bezdek proposed two evaluation indices: partition coefficient ($V_{PC}$) and partition entropy ($V_{PE}$) (*Bezdek, 1974b*; *Bezdek, 1974a*). For both indices, the availability $u_{ij}$ is used as the main criterion for evaluating the clustering quality. For $V_{PC}$, the closer the samples in a cluster are to the cluster center, the closer the value of $u_{ij}$ is to 1. Therefore, the greater the value of $V_{PC}$, the better the clustering quality. $V_{PE}$ is opposite to $V_{PC}$. The maximum value of $u_{ij}$ is 1, and the value of $u_{ij}$ will become 0 after the *log* is taken. For this reason, the value of $V_{PE}$ will not be negative. Therefore, the smaller the value of $V_{PE}$, the better. The disadvantage of $V_{PC}$ and $V_{PE}$ is that they do not consider the spatial structure of clusters, and they are susceptible to the effects of noise and overlap between clusters.

### VH&H

$$V_{H\&H} = \frac{\sum_{i=1}^{K} \sum_{j=1}^{N} u_{ij}^m d^2 \left(x_j, v_i\right)}{N \times \left|\log\min_{s \neq t} d^2\left(v_s, v_t\right) \times w\right|} \qquad (11)$$

$$w = \frac{p_s + p_t}{p_e} = \frac{\sum_{x_j \in C_s} u_{sj} \times \log u_{sj} + \sum_{x_l \in C_t} u_{tl} \times \log u_{tl}}{\sum_{i=1}^{K} \sum_{j=1}^{N} u_{ij} \times \log u_{ij}}. \qquad (12)$$

*Lin, Huang & Huang (2015)* improved the XB index from the perspective of entropy and proposed the index $V_{H\&H}$. The numerator of $V_{H\&H}$ represents the degree of compactness, and the denominator represents the degree of separation. $v_s$ and $v_k$ denote the shortest distance between cluster centers. It can be seen that, for the degree of separation in this index, the *log* is taken to reduce the influence of the shortest distance between cluster centers. Because a negative value may occur after the *log* is taken, it is necessary to take an absolute value. Unlike the XB index, $V_{H\&H}$ introduces the concept of entropy into the degree of separation. According to this characteristic, the *w* in Eq. (12) is used as the weight of the degree of separation. This weight is the result of summing the entropies $p_s$ and $p_k$ in the two clusters with the shortest distance between their centers.

$x_j$ and $x_l$ are samples belonging to clusters $C_s$ and $C_k$, and $u_{sj}$ and $u_{kl}$ denote the degrees of availability of $x_j$ and $x_l$ to $C_s$ and $C_k$. When the clusters $C_s$ and $C_k$ with the shortest distance between their centers overlap each other, the entropies $p_s$ and $p_k$ of the two clusters will be high, *i.e.*, the weight $w$ will be large. Therefore, the degree of separation between the two clusters can be higher. To prevent either $p_s$ or $p_k$ from affecting the final calculation

result excessively, both $p_s$ and $p_k$ need to be normalized. $V_{H\&H}$ has a high tolerance to overlapping between clusters, but it cannot evaluate samples with non-uniform densities stably.

# A NEW CLUSTERING EFFECTIVENESS MEASUREMENT MODEL

The proposed clustering effectiveness measurement model is mainly based on two parts namely, the improved AP clustering algorithm and new internal CVIs. Since the number of clusters produced by the AP clustering algorithm for non-cluster-shaped datasets is higher as compared to the actual number of clusters, the upper bound for clustering, $K_{max}$, is reduced by merging similar clusters, and new internal evaluation indices based on intra-cluster cohesion and inter-cluster dispersion have been designed to address the shortcomings of the commonly used internal evaluation indices mentioned above. The description of the AP clustering algorithm is given in the following subsection.

## Improved AP clustering algorithm

This study adopts the merging idea as described in a study *Gan, Xiuhong & Xiaohui (2015)*. First, the AP clustering algorithm is used to perform the rough clustering of samples. Then, the initial clusters are merged based on similarity, and the upper limit of the number of clusters is reduced to compress the range of clusters and improve clustering accuracy. The general idea is to, on the basis of initial clustering performed by the AP clustering algorithm, calculate the ratio $\alpha$ of the similarity between any two clusters to the average similarity between clusters in the entire sample set. $\alpha$ represents the relationship between any two clusters and the entire sample set in terms of structural similarity. The smaller the value of $\alpha$ is, the closer any two clusters are to each other. If the lowest value of $\alpha$ is within the specified threshold range after complete traversals, the two clusters will be merged. If not, they will endure unmodified. The definitions and equations of the new algorithm are given below.

**Definition 1** The Euclidean distance between any two points in space can be defined as:

$$d\left(x_i, x_j\right) = \sqrt{\sum_{p=1}^{l}\left(x_i^p - x_j^p\right)^2} \tag{13}$$

where $i = 1, 2, \ldots, N$; $j = 1, 2, \ldots, N$; $l$ represents the feature dimensionality of samples.

**Definition 2** The similarity between any two clusters can be defined as taking the sum of the distances between all sample pairs in the two clusters.

$$Sim\left(C_i, C_j\right) = \sum_{x_t \in C_i, x_u \in C_j, i \neq j} d\left(x_t, x_u\right) \tag{14}$$

where $x_t$ and $x_u$ denote any samples in the $i^{\text{th}}$ and $j^{\text{th}}$ clusters, respectively.

**Definition 3** The average similarity between clusters in a sample set can be defined as the average similarity between two clusters in the entire sample set.

$$\overline{Sim(X)} = \frac{\sum_{i=1, j=1, i \neq j}^{K}\left(Sim\left(C_i, C_j\right)\right)}{C_K^2} \tag{15}$$

where $C^2{}_K$ represents the number of combinations of any two clusters arbitrarily selected from $K$ clusters.

**Definition 4** Inter-cluster similarity ratio can be defined as the ratio of the similarity between any two clusters to the average similarity between clusters in the entire sample set.

$$\alpha_{i,j} = Sim(C_i, C_j) / \overline{Sim(X)} \tag{16}$$

$$C_i = \begin{cases} C_i \cup C_j & if\ (\alpha_{i,j} \leq W) \\ C_i & otherwise \end{cases}. \tag{17}$$

If the inter-cluster similarity ratio $\alpha_{ij}$ is less than or equal to a given threshold W, the $i^{th}$ and $j^{th}$ clusters will be merged. Otherwise, they will remain unchanged. The reference range of W used in this paper is [0.3, 0.5]. The specific value of W can be set by users.

## New CVIs

Intra-cluster and inter-cluster similarities are the main elements of the evaluation. Optimal clustering minimizes inter-cluster similarity while maximizing intra-cluster similarity. In this paper, the product of inter-cluster relative density and intra-cluster compactness is used to represent intra-cluster cohesion. The inter-cluster separation is represented by the ratio of the shortest distance between two clusters to the product of the distance between cluster centers and the inter-cluster overlap coefficient.

*Halkidi & Vazirgiannis (2001)* mentioned several indices for the evaluation of clustering algorithms. However, our current study identifies five indices that are particularly suitable for evaluating the Affinity Propagation (AP) algorithm. These indices have demonstrated superior performance in similar studies. For instance, the author stated that intra-cluster cohesion is best suited for unsupervised clustering algorithms (*Estiri, Omran & Murphy, 2018*). The selection of a small set of indices allows for a more focused and in-depth analysis of clustering performance. This enables us to conduct a targeted evaluation of the clustering algorithm's performance, aligning with our primary interest in the current research.

The definitions and equations of the new indices are given below.

### *Intra-cluster cohesion*

From the perspective of intra-cluster similarity, the shorter the distance between data points in the same cluster, the better. Intra-cluster cohesion (hereinafter referred to as coh) is defined as follows:

$$Coh(i) = den(i) \times com(i). \tag{18}$$

The product of inter-cluster relative density (hereinafter referred to as den) and intra-cluster compactness (hereinafter referred to as com) is used to represent intra-cluster cohesion. Specifically, $den(i)$ denotes the inter-cluster relative density of the cluster $C_i$ and is used to measure the ratio of the density of the cluster $C_i$ to that of other clusters, and $com(i)$ denotes the intra-cluster compactness of the cluster $C_i$ and is used to measure the degree of concentration of data points in the cluster. The greater the product of $den(i)$ and

$com(i)$, the greater the value of $Coh(i)$, the higher the intra-cluster cohesion, and the better the clustering result.

### Inter-cluster relative density

The density of low-dimensional samples is usually determined by dividing the number of data points by the occupied area, volume, or space. However, for high-dimensional data, the volume of the data space will increase exponentially, and variations in the measure of distance between any two points in a Euclidean space will become increasingly smaller (*Bezdek, 1974a*; *Lin, Huang & Huang, 2015*). In the structural analysis of a high-dimensional data space, the center of the data space is often "empty." For this reason, the original density calculation method will become invalid. The standard deviation of samples to be clustered can reflect data sparsity. The smaller the standard deviation, the denser the data. The greater the standard deviation, the more discrete the data distribution. Therefore, a new density calculation method is proposed to calculate density by dividing the number of data points in a cluster by the standard deviation of the cluster. The formula for calculating inter-cluster relative density is as follows:

$$den(i) = \frac{n_i/\sigma_i}{(N - n_i)/\left(\sum_{t=1,t\neq i}^{K}\sigma_t/(K-1)\right)} \tag{19}$$

where the numerator represents the density of the cluster $C_i$, $n_i$ is the number of samples in the cluster $C_i$, $\sigma_i$ is the standard deviation of the cluster $C_i$; the denominator represents the average density of other clusters except the cluster $C_i$, and N is the total number of data points in the sample set. A greater value of $den(i)$ means that the density of cluster $C_i$ is higher than the average density of other clusters.

### Intra-cluster compactness

According to the common sense of probability, the probabilities that an approximately normally distributed dataset exists in the confidence intervals $(\mu-\sigma, \mu+\sigma)$, $(\mu-2\sigma, \mu+2\sigma)$, and $(\mu-3\sigma, \mu+3\sigma)$ are 68.26%, 95.44%, and 99.74%, respectively. Considering this property, the ratio between the numbers of samples in different intervals of the same cluster is used to calculate intra-cluster compactness. The equations for calculating intra-cluster compactness are as follows:

$$com(i) = \frac{\sum_{x\in C_i} f(x, v_i, \sigma_i)}{\sum_{x\in C_i} f(x, v_i, 3\sigma_i)} \tag{20}$$

$$num(x, v_i, \sigma_i) = \begin{cases} 1 & d(x, v_i) \leq \sigma_i \\ 0 & otherwise \end{cases} \tag{21}$$

$$\sigma_i = \sqrt{\frac{\sum_{x\in C_i} d(x, v_i)^2}{n_i - 1}} \tag{22}$$

where f(x,vi,σi) is used to measure the similarity or distance at data point x, and cluster center $v_i$ with $\sigma_i$ that represents a scale parameter to adjust the measurement scale. *num* $(x, v_i, \sigma_i)$ denotes the number of data points within a radius of $\sigma_i$ with $v_i$ as the center. The ratio of the number of data points within $\sigma_i$ to that within $3\sigma_i$ is used to represent the degree of concentration of samples in cluster $C_i$. A larger value of this ratio means that more data points in cluster $C_i$ are densely distributed around the cluster center $v_i$. When intra-cluster compactness increases continuously and tends to 1, it means that the data points within the current cluster are extremely similar to each other.

### Inter-cluster dispersion

To minimize the similarity between clusters, the data points in different clusters should be kept as far away from each other as possible. In this paper, the ratio of the shortest distance between two clusters to the product of the distance between cluster centers and the inter-cluster overlap coefficient (hereinafter referred to as overlap) is used to represent inter-cluster dispersion (hereinafter referred to as disp). Inter-cluster dispersion is defined as follows:

$$Disp(i,j) = \frac{\min d\left(x_i, x_j\right)}{d\left(v_i, v_j\right) \times overlap_{ij}} \tag{23}$$

where $x_i$ and $x_j$ denote any samples in clusters $C_i$ and $C_j$, and $v_i$ and $v_j$ denote the centers of the two clusters. The greater the value of this ratio, the more dispersed the clusters, the clearer the cluster boundaries, and the better the clustering quality. In the equation above, the overlap coefficient *overlap* $_{ij}$ represents the degree of concentration of data points in the overlapping area between clusters $C_i$ and $C_j$. The overlap coefficient has several practical applications in a range of research areas. For example, in text document clustering, documents belong to several multiple topics. The overlap coefficient metric is used to analyze the extent to which topics overlap, and potentially reveal the areas of semantic ambiguity. Similarly, the level of ambiguity in object recognition is determined by using the overlap coefficient metric in image segmentation (*Zou et al., 2004*). The overlap coefficient is used in the fractional segmentation of MR images for reproducibility and accuracy.

The overlap coefficient is defined as follows:

$$overlap_{ij} = 1 + \frac{\sum_{x_a \in C_i \cup C_j} f\left(x_a, C_i, C_j\right)}{n_i + n_j} \tag{24}$$

$$f\left(x_a, C_i, C_j\right) = \begin{cases} e^{1 - \left|u_{ia} - u_{ja}\right|} & if \left|u_{ia} - u_{ja}\right| \leq H \\ 0 & otherwise \end{cases} \tag{25}$$

$$u_{ia} = \frac{1}{\sum_{m=1}^{K} \left(d\left(x_a, v_i\right) / d\left(x_a, v_m\right)\right)^2} \tag{26}$$

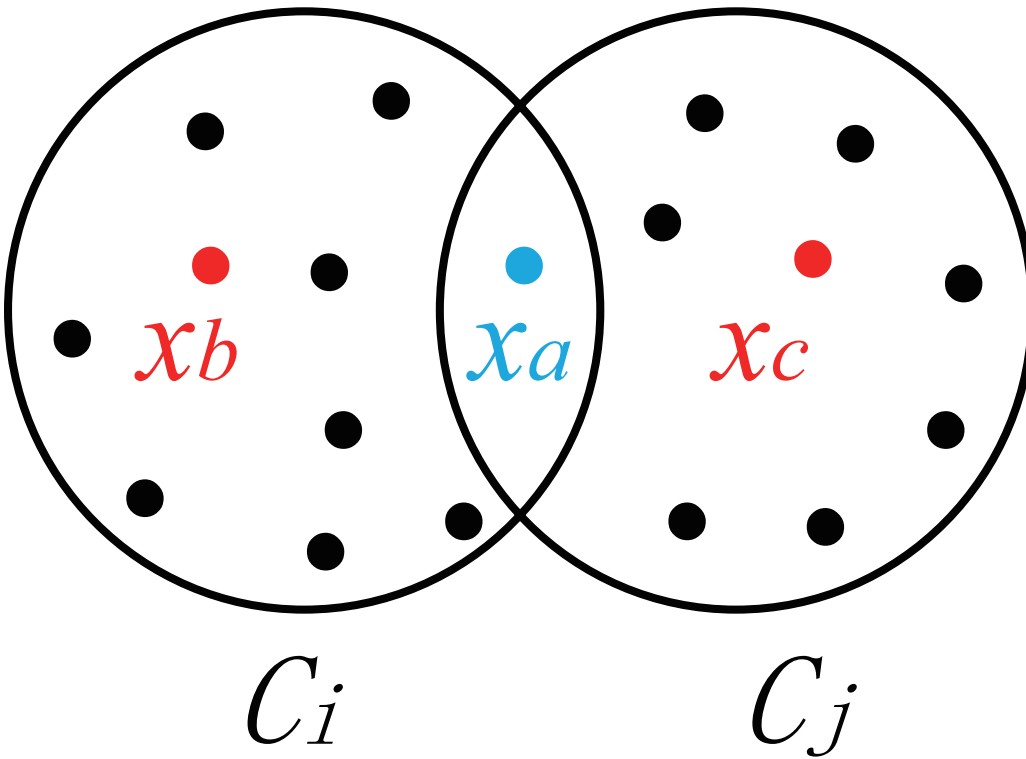

**Figure 1** Degree of dispersion of samples in clusters.

$$u_{ja} = \frac{1}{\sum_{p=m}^{K}\left(d\left(x_a, v_j\right)/d\left(x_a, v_p\right)\right)^2} \tag{27}$$

where $x_a$ denotes the data in the cluster $C_i$ or cluster $C_j$; $n_i$ and $n_j$ denote the number of data points in $C_i$ and the number of data points in $C_j$; $f(x_a, C_i, C_j)$ represents the contribution of the sample $x_a$ to the degree of concentration in the overlapping area between clusters $C_i$ and $C_j$; $u_{ia}$ and $u_{ja}$ are the degrees of availability of $x_a$ to clusters $C_i$ and $C_j$ (*Dunnâ, 1974*); and H is the threshold at which $x_a$ falls within the overlapping area. The default value of H used in this paper is 0.3. As shown in Fig. 1, $x_b$ and $x_c$ are located inside $C_i$ and $C_j$, respectively, $x_b \in C_i$, and $x_c \in C_j$. The availability of $x_b$ and $x_c$ is very clear, *i.e.*, $|u_{ib} - u_{jb}| >$H, $|u_{ic} - u_{jc}| >$H. Compared with the availability of $x_b$ and $x_c$, the availability of the data point $x_a$ located in the overlapping area between the two clusters (as shown in Fig. 1) is closer to $u_{ia}$ and $u_{ja}$, *i.e.*, $|u_{ia} - u_{ja}| \leq$H. $f(x_a, C_i, C_j)$ determines the inter-cluster overlap coefficient by calculating the degree of concentration of fuzzy data points in the overlapping area.

Specifically, when $|u_{ia} - u_{ja}| \leq$ H, data points will fall within the overlapping area. In this case, $e^{1-|u_{ia}-u_{ja}|}$ is taken to increase the influence of data points in the overlapping area. Otherwise, zero is taken. From Eqs. (24) and (25), it can be known that the smaller the

cumulative sum of the degrees of concentration in the overlapping areas between clusters, the smaller the overlap coefficient, and the clearer the division of clusters.

### New internal evaluation indices

The new internal evaluation indices introduced herein include intra-cluster cohesion and inter-cluster dispersion. They are used to determine the optimal number of clusters for the final *CVI*. This evaluation index is defined as follows:

$$CVI(K) = \frac{1}{C_K^2} \sum_{i=1}^{K} \sum_{j=1, j \neq i}^{K} Coh(i) \times Coh(j) \times Disp(i,j) \qquad (28)$$

where $C^2{}_k$ is the number of combinations of any two clusters arbitrarily selected from $K$ clusters, $Coh(i)$ and $Coh(j)$ denote the intra-cluster cohesion of the cluster $C_i$ and that of the cluster $C_j$, $Disp(i,j)$ denotes the inter-cluster dispersion between clusters $C_i$ and $C_j$. Both *Coh* and *Disp* are positively correlated with the quality of clustering. Therefore, the evaluation index $CVI(K)$ will achieve optimal results when the product of the intra-cluster cohesion and inter-cluster dispersion of $C_i$ and $C_j$ reaches the maximum value.

### The optimal number of clusters

Based on the nature of the *CVI*, it is evident that a higher index value indicates better clustering quality, except for the Xie-Beni (XB) index, which achieves improved clustering quality when the index reaches its lowest possible value. However, Therefore, the optimal number of clusters $K_{opt}$ is the number of clusters when $CVI(K)$ reaches the maximum value.

$$K_{opt} = \arg \max \{CVI(K)\} \qquad (29)$$

where $K \in [2, K_{max}]$, and $K_{\max}$ are given by the improved AP clustering algorithm proposed herein.

### Clustering effectiveness measurement model based on merging similar clusters

In this article, the original AP clustering algorithm is improved by merging similar clusters, and a clustering effectiveness measurement model based on merging similar clusters (hereinafter referred to as the CEMS) is proposed based on the new evaluation indices presented herein. The structure of the CEMS is shown in Fig. 2.

The operation process of the CEMS consists of 10 steps intended to merge similar clusters and determine the optimal number of clusters. These steps are described below.

(1) Calculate the similarity matrix $S$ of dataset $X$ using Eq. (1).

(2) Calculate and update the responsibility and availability using Eqs. (2) and (3), set the damping parameter and the number of iterations, and use Eqs. (4) and (5) to correct the responsibility and availability during iterations and reduce the amplitude of oscillation.

(3) Select the data point with the maximum sum of responsibility $r$ and availability $a$ according to Eq. (6), set the selected data point as the representative point of the cluster to which it belongs, repeat Step (2), obtain the representative points of all clusters, and complete the clustering of non-representative points based on similarity.

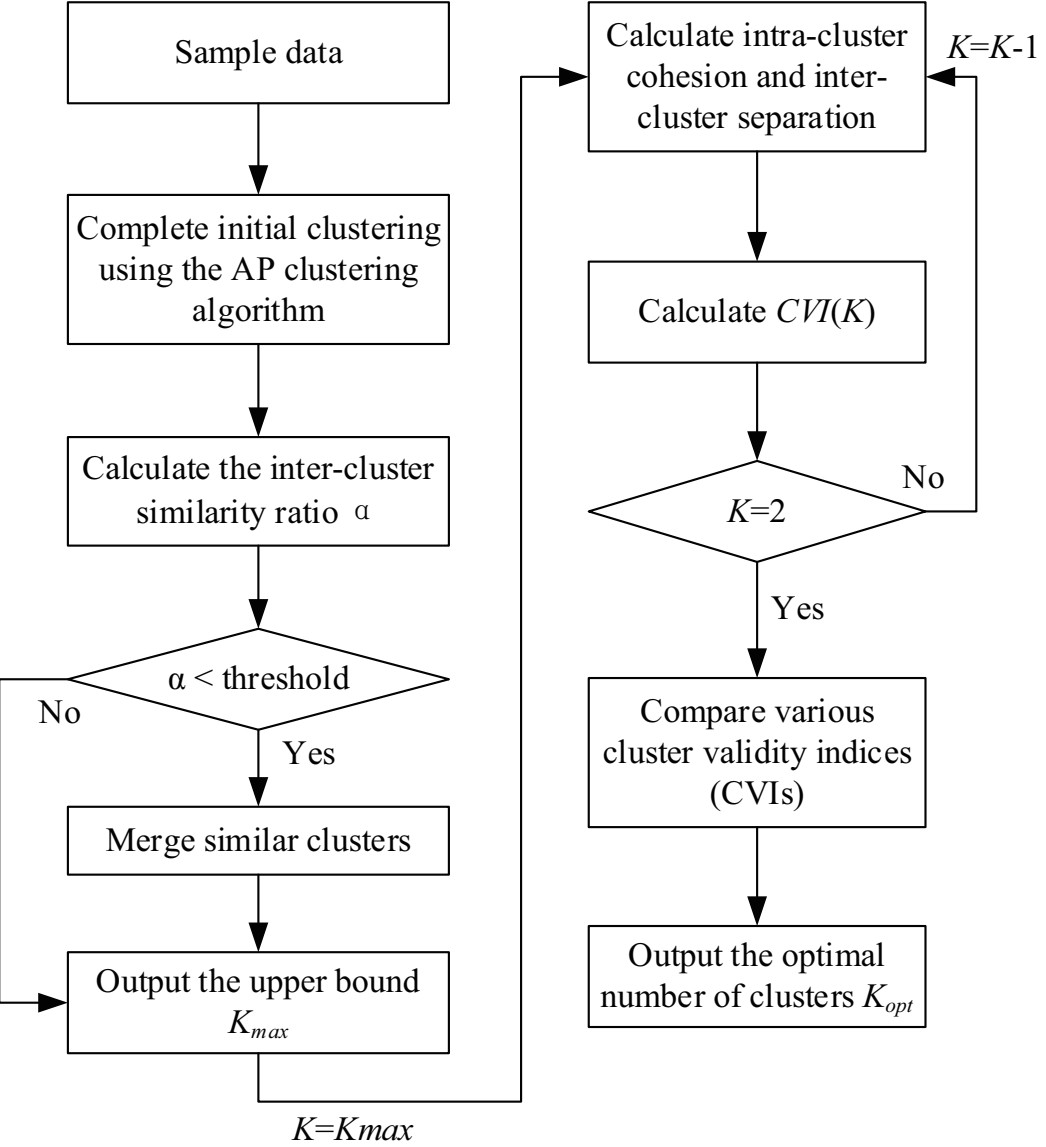

**Figure 2   CEMS process flowchart.**

(4) Calculate the similarity between any two clusters and the average similarity between clusters in the entire sample set using Eqs. (14) and (15).

(5) Calculate the inter-cluster similarity ratio using Eq. (16) and use Eq. (17) to merge clusters that satisfy the required condition.

(6) Repeat steps (4) and (5), update clusters based on the new inter-cluster similarity ratio, and obtain the maximum number of clusters $K_{max}$ at the end of the iterative process.

(7) Take $K_{max}$ as the initial value of the parameter $K$ in the $CVI$, i.e., $K = K_{max}$.

(8) Calculate inter-cluster relative density ($den$), intra-cluster compactness ($com$), and inter-cluster overlap coefficient ($overlap$) using Eqs. (19), (20), and (24), calculate

**Table 1  UCI dataset.**

| Dataset | Sample size | Number of features | Standard number of clusters |
|---|---|---|---|
| Iris | 150 | 4 | 3 |
| Wine | 178 | 13 | 3 |
| Thyroid | 215 | 5 | 2 |
| Yeast | 1,484 | 8 | 10 |
| Glass | 214 | 9 | 6 |
| Dermatology | 366 | 34 | 6 |

intra-cluster cohesion (*coh*) and inter-cluster dispersion (*disp*) using Eqs. (18) and (23), and calculate the *CVI* using Eq. (28).

(9) Let $K = K$-1 and repeat steps (7) and (8) until $K = 2$ to obtain a set of *CVI* values, *i.e.*, $CVI\_K = \{CVI(2), CVI(3),\ldots, CVI(K_{max}-1), CVI(K_{max})\}$.

(10) Select the parameter $K$ corresponding to the maximum value of *CVI* according to Eq. (29), and output it as the optimal number of clusters $K_{opt}$.

## EXPERIMENTS AND RESULTS

The experimental environment is as follows: Intel(R) Core(TM) i7-10750H CPU @2.6 GHz 2.59 GHz, 16 GB RAM, Microsoft Windows 10 (64-bit) Professional, and the test platform is Matlab (*Liang & Cheng, 2020*). The experiments conducted include the CEMS effectiveness test and practicality test. Specifically, the first experiment is designed to perform the comparative test of the number of clusters using the improved AP clustering algorithm in combination with the *CVI* and commonly used internal evaluation indices. Its purpose is to verify the effectiveness of the improved AP clustering algorithm and the *CVI*. The details of the datasets used are listed in Table 1.

The second experiment is designed to further verify the functional integrity of the CEMS from the perspectives of intrusion detection rate, correct classification rate, FNR, and FPR using the NSL-KDD dataset (*Liu, Lu & Zhang, 2020*). The intrusion detection rate is used to measure the actual proportion of intrusions that are correctly identified by the proposed method. Correct classification rate and accuracy are interchangeably used. Accuracy measures the overall proportion of correctly classified instances. False negative rate (FNR) is used to measure the proportion of actual intrusions that are incorrectly classified, while false positive rate (FPR) measures those non-intrusion instances that have been incorrectly classified as intrusions.

The NSL-KDD dataset is structured into distinct categories based on network traffic types. The NSL-KDD dataset is widely employed to evaluate the effectiveness of intrusion detection systems. This dataset does not include duplicate records. Each record in this dataset possesses 42 attributes: 41 attributes are about the characteristics of the dataset and one attribute represents the type of attack. This data is owned by the Canadian Institute for Cybersecurity and can be accessed from "https://www.unb.ca/cic/datasets/nsl.html". Test sets (T1, T2, and T3) are employed for CEMS testing. For example, in T1, there are

**Table 2  Comparison of the numbers of clusters for original AP and improved AP algorithms.**

|  | Standard number of clusters | Original AP clustering algorithm | Improved AP clustering algorithm |
|---|---|---|---|
| Iris | 3 | 8 | 6 |
| Wine | 3 | 9 | 6 |
| Thyroid | 2 | 7 | 4 |
| Yeast | 10 | 23 | 18 |
| Glass | 6 | 20 | 15 |
| Dermatology | 6 | 17 | 12 |

**Table 3  Comparison of the numbers of clusters for internal evaluation indices.**

|  | Standard number of clusters | $DB.$ | $XB$ | $V_C$ | $V_{PE}$ | $V_{H\&H}$ | $CVI$ |
|---|---|---|---|---|---|---|---|
| Iris | 3 | 2 | 2 | 3 | 3 | 3 | 3 |
| Wine | 3 | 3 | 9 | 4 | 3 | 3 | 3 |
| Thyroid | 2 | 5 | 8 | 2 | 2 | 2 | 2 |
| Yeast | 10 | 10 | 10 | 8 | 7 | 10 | 10 |
| Glass | 6 | 2 | 8 | 2 | 2 | 15 | 6 |
| Dermatology | 6 | 4 | 4 | 2 | 2 | 14 | 5 |

3,070 "Normal" connections, 308 instances of "Dos" (Denial-of-Service), 234 "Probe" instances, 121 "R2L" instances (unauthorized access from a remote machine), and 42 "U2R" instances (unauthorized access to root). Similarly, values for the T2 and T3 test sets are also provided.

## CEMS effectiveness test

In this section, a comparative experiment on the maximum number of clusters for the UCI dataset was conducted using the improved AP clustering algorithm and the original AP clustering algorithm. The settings of these two algorithms are as follows: the damping factor $\lambda = 0.85$, the number of iterations $t = 1,500$, and the threshold of inter-cluster similarity ratio for the improved AP clustering algorithm $W = 0.3$. The results are given in Table 2. The clustering results were evaluated using the $CVI$ and classical internal evaluation indices. The comparison results are summarized in Table 3.

As shown in Table 2, the values of upper bounds for the improved AP clustering algorithm on the UCI dataset are all smaller than those for the original AP clustering algorithm, indicating that the former can effectively compress the clustering space.

As shown in Table 3, the accuracy of the $CVI$ concerning the number of clusters is 83.33%, and the accuracy levels of other indices are 33.33%, 16.67%, 33.33%, 50%, and 66.67%, respectively. It is obvious that the accuracy of the $CVI$ is significantly higher than the accuracy levels of the other five indices. It is to be noted that during the test on the Glass dataset, all indices except the $CVI$ failed to achieve accurate classification. During the test on the Dermatology dataset, all indices failed to achieve accurate classification, and the $CVI$ divided the data points in this dataset into five clusters. Compared with the

**Table 4  NSL-KDD dataset.**

| Data type | Training set | Test set T1 | Test set T2 | Test set T3 |
|---|---|---|---|---|
| Normal | 19,106 | 3,070 | 3,025 | 3,052 |
| Dos | 5,180 | 308 | 321 | 315 |
| Probe | 612 | 234 | 250 | 220 |
| R2L | 258 | 121 | 111 | 105 |
| U2R | 40 | 42 | 52 | 46 |

number of clusters determined by other indices, the number of clusters determined by the *CVI* is closer to the standard number of clusters. In general, the *CVI* performs best on this dataset. It indicated that *CVI* better performed on the high dimensional datasets due to the thorough exploitation of spectral dimension reduction, optimization of data architecture, and reduction in the overlapping between various clusters. As a result of it, the clustering performance is improved.

## Application of the CEMS in intrusion detection

To test the performance of the CEMS, 20% of the KDDTrain+ samples in the NSL-KDD intrusion detection dataset were experimentally selected to create a training set with 25,196 samples, and 50% of the KDDTest+ samples were selected and divided into three test sets with 11,272 samples in total. The samples in the training and test sets contain four network intrusions, namely, Dos, Probe, R2L, and U2R (*Chen & Wang, 2022*). The formats of the samples were converted by one-hot encoding (*Zhang et al., 2019*), and the feature dimensionality of the datasets was reduced using the dimensionality reduction method. The details of pre-processed data are listed in Table 4.

## Relationship between the *CVI* and K

The improved AP clustering algorithm was run with the parameters listed above. The maximum number of clusters in training is set as $K_{max} = 28$, denoted along the $x$-axis. The *CVI* values corresponding to different values of $K$ are shown in Fig. 3, denoted along the $y$-axis. When $2 \leq K \leq 20$, the *CVI* value increases continuously. When $K \in$ as mentioned in *Li et al. (2017)*, the increase in the *CVI* value slows down. When $K = 24$, the *CVI* value reaches its peak.

As the value of $K$ continues to increase, the *CVI* value decreases slowly. When $K = 27$, the rate of decrease in the *CVI* value increases. Considering the rules of variation in the *CVI*, the set of multiple numbers of clusters corresponding to the process of the *CVI* slowly increasing to the maximum value and then slowly decreasing from the maximum value is defined as the optimal clustering space denoted as $K_{opt} \in$ as mentioned in *Davies & Bouldin (1979)*.

## Indices for intrusion detection in the optimal clustering space

In this section, the effectiveness of the indices for intrusion detection in the optimal clustering space $K_{opt} \in$ was verified using three test sets (*Davies & Bouldin, 1979*).

Table 5 presents various indices for three test sets (T1–T3). Clusters 20–26 are provided with their corresponding detection rate, correct classification rate, false negative rate (FNR),

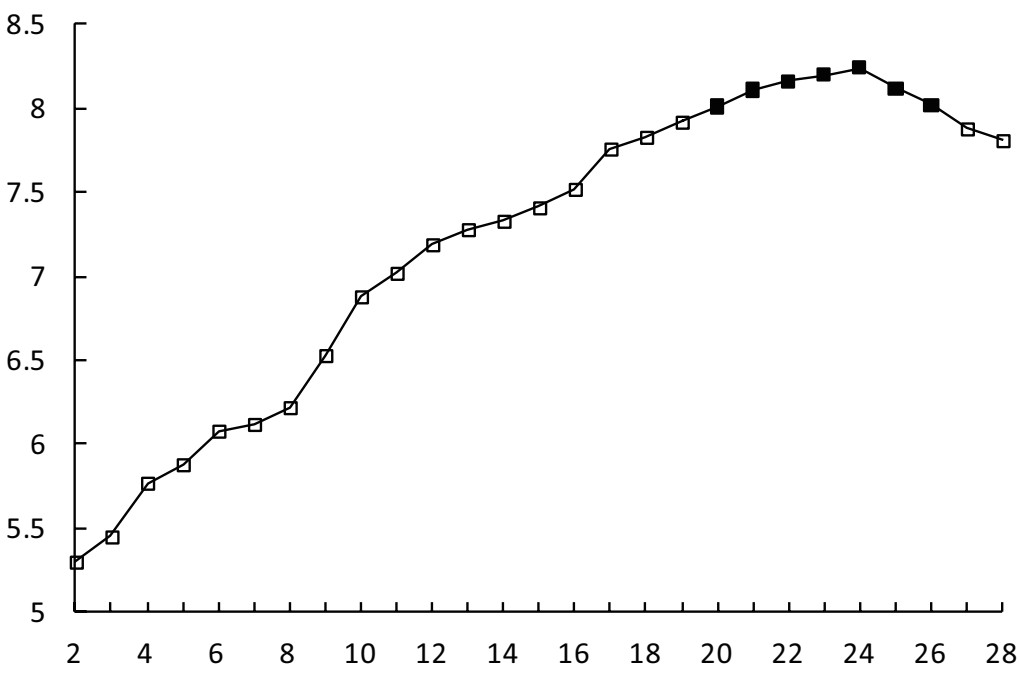

**Figure 3  CVI-K relationship.**

**Table 5  Test results of various indices for three test sets ($K_{opt} \in [13, 19]$).**

| K | Detection rate | | | Correct classification rate | | | FNR | | | FPR | | |
|---|---|---|---|---|---|---|---|---|---|---|---|---|
| | T1 | T2 | T3 | T1 | T2 | T3 | T1 | T2 | T3 | T1 | T2 | T3 |
| 20 | 93.82 | 92.87 | 93.81 | 92.13 | 92.25 | 92.87 | 5.26 | 5.27 | 5.3 | 5.35 | 5.26 | 5.43 |
| 21 | 94.11 | 93.33 | 94.03 | 92.67 | 93.29 | 93.08 | 4.98 | 4.93 | 4.98 | 4.16 | 4.28 | 4.64 |
| 22 | 94.45 | 93.64 | 94.29 | 93.03 | 93.02 | 94.16 | 4.83 | 4.71 | 4.73 | 4.65 | 4.71 | 4.13 |
| 23 | 93.88 | 92.58 | 93.85 | 93.42 | 92.25 | 93.43 | 5.55 | 5.08 | 5.23 | 4.92 | 5.39 | 4.85 |
| 24 | 93.41 | 92.31 | 93.41 | 93.11 | 92.14 | 93.18 | 6.23 | 5.99 | 5.73 | 5.12 | 4.94 | 5.04 |
| 25 | 93.33 | 91.83 | 93.22 | 92.52 | 91.57 | 92.51 | 6.88 | 6.89 | 6.68 | 5.26 | 5.33 | 5.33 |
| 26 | 93.82 | 91.64 | 92.69 | 92.61 | 91.48 | 92.11 | 6.96 | 6.83 | 6.93 | 5.03 | 4.94 | 5.58 |

and false positive rate (FPR) values. From the values listed in Table 5 and the line charts in Fig. 4, it can be seen that when $K = 22$, the detection rates and FNR of the three test sets reach extreme values at the same time, and the average detection rate and FNR are 94.13% and 4.76%, respectively; when $21 \leq K \leq 23$, the correct classification rates of the test sets reach the maximum values successively, with an average of 93.40%; when $K = 21$ and $K = 22$, the FPR of the three test sets reach the minimum values, with an average of 4.36%.

## Time complexity analysis

The AP algorithm exhibits quadratic complexity, meaning that as the size of a dataset or the number of data points increases, the computational demand of the AP algorithm grows quadratically. Performance peaks of the algorithm were observed at a specific

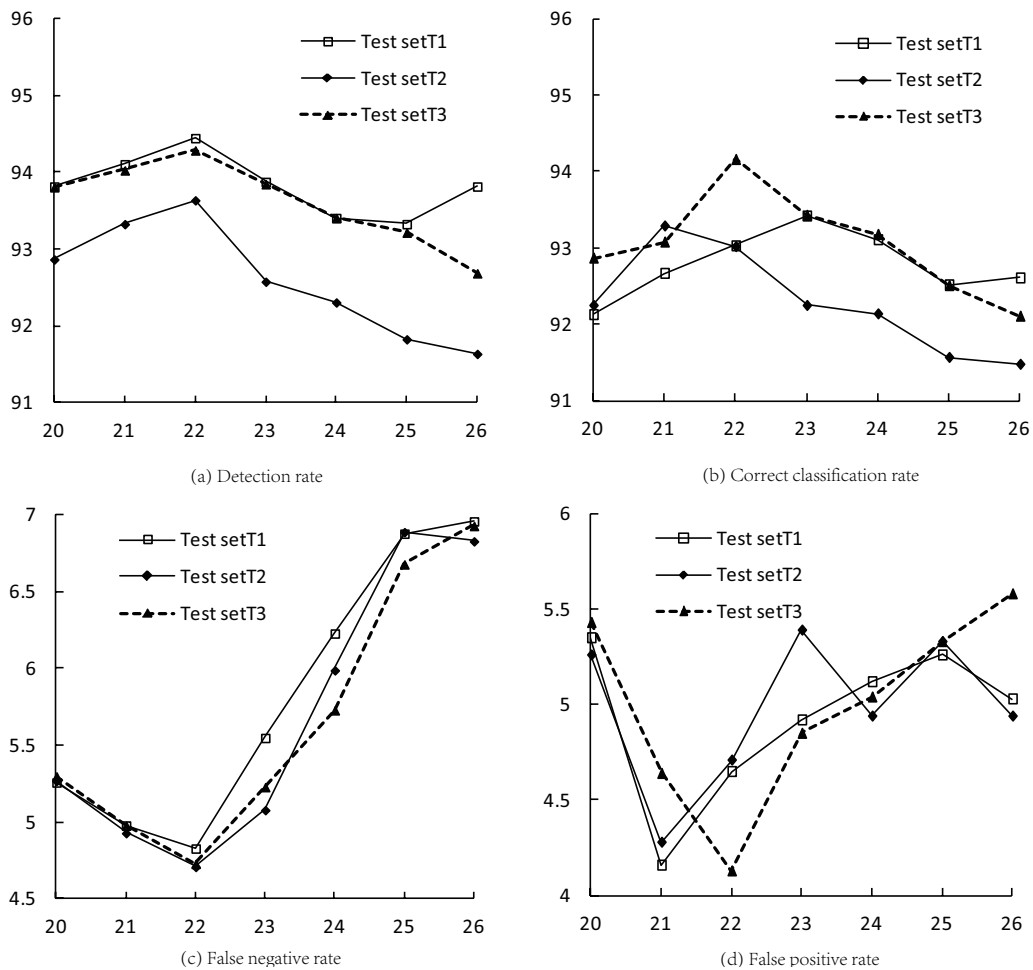

**Figure 4** **Indices for intrusion detection in the optimal clustering space.**

cluster ($K = 22$), indicating that as the number of clusters increases, the computational requirements of the AP algorithm also escalate. Therefore, due to the quadratic time complexity of the AP algorithm, it is crucial to carefully consider its application, especially with large datasets.

## Comparison between the CEMS and other algorithms

The performance comparison of CEMS and other algorithms is presented in Tables 6 to 9. Specifically, Table 6 showcases the comparison of detection rates (*Mohammadi et al., 2019*; *Su et al., 2020*), while Table 7 displays the comparison of correct classification rates (*Zou & Yang, 2018a*; *Zou & Yang, 2018b*).

For further verification of the *CVI* and compare the clustering quality of the CEMS and that of other algorithms horizontally, the K-medoids clustering algorithm and two improved density-based clustering algorithms (*Zou & Yang, 2018a*; *Zou & Yang, 2018b*) were combined with the *CVI* and recorded as "K-medoids+," "Reference (*Zou & Yang, 2018a*)+," and "Reference (*Zou & Yang, 2018b*)+." Four intrusion detection indices were

**Table 6  Results of comparison between the detection rates (%) of various clustering algorithms.**

| K | K-medoids+ | Reference [33]+ | Reference [34]+ | CEMS |
|----|-----------|-----------------|-----------------|-------|
| 20 | 91.49 | 92.73 | 92.81 | 93.50 |
| 21 | 91.86 | 93.46 | 93.52 | 93.82 |
| 22 | 92.64 | 93.13 | 93.63 | 94.13 |
| 23 | 92.35 | 92.75 | 93.26 | 93.44 |
| 24 | 92.07 | 92.62 | 92.83 | 93.04 |
| 25 | 91.85 | 92.33 | 92.64 | 92.79 |
| 26 | 91.59 | 91.69 | 91.91 | 92.72 |

**Table 7  Results of comparison between the correct classification rates (%) of various clustering algorithms.**

| K | K-medoids+ | Reference [35]+ | Reference [36]+ | CEMS |
|----|-----------|-----------------|-----------------|-------|
| 20 | 90.50 | 92.03 | 92.47 | 92.42 |
| 21 | 91.43 | 92.31 | 92.62 | 93.01 |
| 22 | 91.88 | 92.26 | 92.88 | 93.40 |
| 23 | 91.42 | 91.86 | 93.05 | 93.03 |
| 24 | 91.13 | 91.62 | 92.41 | 92.81 |
| 25 | 90.79 | 91.27 | 92.22 | 92.20 |
| 26 | 90.62 | 90.70 | 91.46 | 92.07 |

horizontally compared within the range of $K_{opt} \in$ as mentioned in *Davies & Bouldin (1979)*. The k-medoids algorithm attempts to minimize the distance between two points and its centroid. For the K-medoids+ algorithm, the average value was taken after it was run 500 times.

By observing the data in Tables 6 and 7, it can be found that, after the other three algorithms are combined with the *CVI*, the detection rate, as well as the correct classification rate, can receive their highest values when $21 \leq K \leq 23$, and the detection rate of the CEMS is significantly higher than those of the other three algorithms within the whole range given above. The correct classification rate of the CEMS is consistent with that of the algorithm proposed in *Zou & Yang (2018b)* when $K = 23$ and the CEMS outperforms the other three algorithms in the ranges of $20 \leq K$ 23 and 25x $K \leq 26$. As shown in Tables 6 and 7 the proposed CEMS achieved the best performance in terms of detection rate and accuracy (%) on the NSL-KDD dataset. Results clearly show that the detection and classification performances of the proposed CEMS are more effective compared with the rest of the clustering algorithms.

From the data in Table 8, it can be seen that the FNR of the other three clustering algorithms can reach the minimum values when $21 \leq K \leq 23$, and the minimum FNR of the CEMS is 4.76%, which is slightly lower than that of "Reference *Zou & Yang (2018b)*+" and significantly lower than those of the other two algorithms. The performance of the CEMS is better compared with the K-medoid, and clustering models in Reference *Zou & Yang (2018a)* and Reference *Zou & Yang (2018b)*. False negatives are either due to poor detection probabilities or failure in connecting clusters that are too far apart. However, in

**Table 8** Results of comparison between the false negative rates (%) of various clustering algorithms.

| K | K-medoids+ | Reference [35]+ | Reference [36]+ | CEMS |
|---|---|---|---|---|
| 20 | 8.33 | 7.23 | 5.33 | 5.28 |
| 21 | 7.96 | 6.86 | 5.12 | 4.96 |
| 22 | 7.62 | 7.22 | 4.97 | 4.76 |
| 23 | 7.58 | 7.59 | 5.56 | 5.29 |
| 24 | 7.87 | 7.87 | 6.11 | 5.98 |
| 25 | 7.83 | 8.13 | 6.94 | 6.82 |
| 26 | 8.67 | 8.47 | 7.11 | 6.91 |

**Table 9** Results of comparison between the false positive rates (%) of various clustering algorithms.

| K | K-medoids+ | Reference [35]+ | Reference [36]+ | CEMS |
|---|---|---|---|---|
| 20 | 5.39 | 5.52 | 5.64 | 5.35 |
| 21 | 4.48 | 4.74 | 4.78 | 4.36 |
| 22 | 4.57 | 4.63 | 4.68 | 4.5 |
| 23 | 5.13 | 5.09 | 5.21 | 5.05 |
| 24 | 4.95 | 5.06 | 5.03 | 5.03 |
| 25 | 5.21 | 5.26 | 5.16 | 5.31 |
| 26 | 5.35 | 5.19 | 5.22 | 5.18 |

the current study, the proposed CEMS's reduced FNR indicates its robust and efficacious performance.

By observing the data in Table 9, it can be known that the FPR of the other three algorithms can reach the minimum values when $21 \leq K \leq 22$; when $20 \leq K \leq 23$, the FPR of the CEMS is slightly lower than those of the other three algorithms; and when $24 \leq K \leq 26$, the average FPR of the CEMS is basically consistent with those of "K-medoids+" and "Reference *Zou & Yang (2018a)*+" and slightly higher than that of "Reference *Zou & Yang (2018b)*+".

The experimental results above show that the *CVI* has a high universality and can accurately assess the results from clustering and give an optimal number of clusters after it is combined with other clustering algorithms. During the comparative test, the CEMS could improve the intrusion detection rate and correct classification rate at a low FPR and significantly reduce the FNR, indicating that its overall performance is more desirable than that of the other three clustering algorithms.

The current research shows promising results on both the UCI and NSL-KDD datasets. This suggests that the findings of this study may be generalized to other datasets with similar characteristics. However, the applicability of the proposed research may be limited if new datasets have different features.

This research show cases correct classification and clustering, demonstrating that the proposed approach was effectively applied. The use of internal evaluation indices has further strengthened confidence in the internal validity of this study. It demonstrates that the observed effects and their association are genuinely attributed to the performance

of the CEMS algorithm. This implies that the proposed research was well-designed and conducted with minimized potential for biases in the experimental design.

## CONCLUSION

In this article, an improved AP clustering algorithm is proposed, which uses the ratio of the similarity between any two clusters to the average similarity between clusters in the entire sample set as a reference, reduces the upper limit of the number of clusters by merging similar clusters, and solves some problems with the original AP clustering algorithm, such as excessively large number of clusters and low accuracy. In addition, several new internal evaluation indices are proposed, the product of inter-cluster relative density and intra-cluster compactness is used to improve intra-cluster cohesion, and the inter-cluster overlap coefficient is used to enhance inter-cluster separation and mitigate the effects of uneven distribution of data points and overlap between clusters. Internal evaluation indices show a better solution to issues with classical ones: monotony and excessive clusters. The findings show that CEMS is practical and versatile as CVIs enhance accuracy with other algorithms.

In future work, we will set thresholds for parameters such as the bias parameter $p$, the inter-cluster similarity ratio $w$, and the inter-cluster overlap coefficient $H$, increase the correct classification rate, and seek a better balance between high detection rate and low FPR to improve the CEMS.

Since lower FPR is desirable, we further need to conduct more experiments to achieve the highest accuracy with just over 1% false positives. In future works, we plan to conduct an extensive study of clustering algorithms to give an enhanced detection and classification solution by using real-time datasets.

The proposed method can be extended by applying a post-processing step to merge clusters based on a similarity threshold. Using techniques such as agglomerative clustering with the suitable linkage method to combine clusters with sufficient similarity. Further, a hierarchical approach can be combined with the AP algorithm which can give us more control over the number of resulting clusters, allowing us to merge similar clusters at different hierarchy levels.

### Funding

This research was supported by the Educational Science Planning Project of Guangdong Province (2022GXJK490), the Scientific Research Projects of the Department of Education of Guangdong Province (2021KTSCX227, 2021ZDZX1124), and the projects of Shaoguan Science and Technology Bureau (200811224533986, 210718114531595). The funders had no role in study design, data collection and analysis, decision to publish, or preparation of the manuscript.

### Grant Disclosures

The following grant information was disclosed by the authors:

The Educational Science Planning Project of Guangdong Province: 2022GXJK490.
The Scientific Research Projects of the Department of Education of Guangdong Province: 2021KTSCX227, 2021ZDZX1124.
The projects of Shaoguan Science and Technology Bureau: 200811224533986, 210718114531595.

## Competing Interests

The authors declare there are no competing interests.

## Author Contributions

- Guiqin Duan performed the experiments, performed the computation work, prepared figures and/or tables, and approved the final draft.
- Chensong Zou conceived and designed the experiments, analyzed the data, authored or reviewed drafts of the article, and approved the final draft.

## Data Availability

The NSL-KDD dataset is available in the Supplementary Files.

The Third-Party Data is available at:

- Iris: http://archive.ics.uci.edu/dataset/53/iris
- Wine: http://archive.ics.uci.edu/dataset/109/wine
- Thyroid: http://archive.ics.uci.edu/dataset/102/thyroid+disease
- Yeast: http://archive.ics.uci.edu/dataset/110/yeast
- Glass: http://archive.ics.uci.edu/dataset/42/glass+identification
- Dermatology: http://archive.ics.uci.edu/dataset/33/dermatology
- NSL-KDD: https://www.unb.ca/cic/datasets/nsl.html

## Supplemental Information

Supplemental information for this article can be found online at http://dx.doi.org/10.7717/peerj-cs.1863#supplemental-information.

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
