# Peer review of "A clustering effectiveness measurement model based on merging similar clusters"

_PeerJ Computer Science, doi:10.7717/peerj-cs.1863_

## Round 0.1 · original submission · Major Revisions

I have received the review reports for your paper submitted to PeerJ Computer Science from the reviewers. According to the reports, I will recommend major revision to your paper. Please refer to the reviewers’ opinions to improve your paper. Please also write a revision note such that the reviewers can easily check whether their comments are fully addressed. Note that a revision does not guarantee the final acceptance of your paper. You must convince the reviewers as possible as you can. We look forward to receiving your revised manuscript soon.

Reviewer 1 ·

Basic reporting

The main idea of this paper is to propose a clustering effectiveness measurement model based on merging similar clusters to address the problems experienced by the affinity propagation (AP) algorithm in the clustering process, such as excessive local clustering, low accuracy, and invalid clustering evaluation results that occur due to the lack of variety in some internal evaluation indices when the proportion of clusters is very high. The model reduces the upper limit of the number of clusters by merging similar clusters, and new internal evaluation indices based on intra-cluster cohesion and inter-cluster dispersion have been designed to address the shortcomings of the commonly used internal evaluation indices. The model has been comprehensively tested on UCI and NSL-KDD datasets, and evaluated and compared using indicators such as detection rate, correct classification rate, FNR, FPR, etc. The results show that the model has advantages in clustering accuracy and efficiency. I have the following suggestions:
1. There are some errors in the definition of the equation, such as the function definition f(xa,ci,ci) of the 25th equation should be f(xa,ci,cj), and the vm in the denominator part d(xa,vm) of the 27th equation should be vp.
2. Since f(x,vi,σi) and f(xa,ci,cj) express different functions, it is recommended to use different function names.
3. Missing commas in function f(x,viσi) on line 328.
4. As for the overlap coefficient overlapij, it is suggested to use examples to analyze its rationality and show it in the form of a graph.
5. It is recommended to give the analysis of time complexity.

Experimental design

1, In the experiment, it is suggested to give the code of the experiment to ensure the repeatability of the algorithm.
2. it is suggested to give a detailed explanation of relevant indicators, such as intrusion detection rate, correct classification rate, FNR, and FPR.
3. please explaine the result of Table 5. What is the meaning of K, T1, T2, T3?

Validity of the findings

no comment

Additional comments

no comment

Cite this review as

Reviewer 2 ·

Basic reporting

The topic of the article is important, because there is no universal clustering algorithm. Many of them are able to detect clusters of a particular shape. One of the main problems in solving the clustering problem is the problem of determining the optimal number of clusters. Different authors offer new approaches to solving the clustering problem and new methods for determining the optimal number of clusters.
In this article, the authors attempted to modify the affinity propagation (AP) algorithm and proposed the clustering effectiveness measurement model based on merging similar clusters.

First, the authors perform a review of known indices used in choosing the optimal number of clusters. However, there is a question why these 5 indices were chosen, although in other works of the authors mentioned in the article under consideration (for example, [12]) a large list of indices is presented.
The following article is worth considering:
Maria Halkidi, Yannis Batistakis, Michalis Vazirgiannis. On Clustering Validation Techniques. Journal of Intelligent Information Systems. Vol. 17, Issue 2-3, December 2001, pp 107–145, https://doi.org/10.1023/A:1012801612483
(https://web.itu.edu.tr/sgunduz/courses/verimaden/paper/validity_survey.pdf)

A large number of indexes can be found in the works of other authors.
The authors work with a crisp clustering algorithm. In this regard, it is not clear why 4 out of 5 indexes involve working with fuzzy clustering algorithms. Where do the authors in this case take the values for the membership functions, which can be obtained using fuzzy clustering algorithms, for example, fuzzy c-means. How, then, were the scores given in the tables obtained?
This is a key question, without an answer to which it is problematic to evaluate the results of the experiments.

At the same time, there is another question: why do the authors not consider the cluster silhouette index, which is now actively used?

The following aspects of the article are questionable.
1. In the "Internal evaluation indices" section, the authors mention 5 indices, but do not provide a description of the parameters used in the formulas.
In some formulas the number of clusters is denoted by a small letter k, and in others, by a capital letter K.
Readers will have to immediately open the articles on the suggested links and try to understand what x, vi, d, ni, k, Ci, K, N, m, uij and w are.
Obviously, all parameters must be described and unified for the current article, so that there are no discrepancies (so that the same parameter is not designated differently).

The authors try to describe some parameters only when describing the VH&H index.
uij is the fuzzy degree of belonging of the i-th object to the j-th cluster, but the authors never use fuzziness anywhere!
There is also a note about indexes (and some other symbols): if italic is used, then it should be used for all indexes.
It is not great when indexes i, j are used for indexing both objects and clusters. It is also not great that index k is used to index clusters, although it is also used to indicate the number of clusters.

2. The authors state in lines 377-380 that all cluster validity indices (CVIs) should be maximized: "From the nature of the CVI, it can be known that the larger the value of this index, the better the clustering quality".
Perhaps this statement only applies to the proposed index, since, for example, the XB index must be maximized. At the same time, there are recommendations on what values of this index indicate a good quality of clustering.

3. In formula (28) is the square of the number Сk in the denominator or is it such a strange designation (judging from the context)?

4. Line 436 contains the numbers "33.33%, 16.67%, 33.33%, 50%, and 66.67%", i.e. 4 fuzzy clustering indices are involved in the work. But where is this fuzzy clustering?

5. Many lines in the experimental part contain constructions that cause confusion, for example, in line 459: "K∈[14]". Square brackets are usually used to indicate a link to source from the list of references. Such constructions also appear in lines 459, 464, 467, 478.

6. There is a typo in line 492: "Reference 28]+". A space is missing on line 499: "the average FPRof".

7. The names of tables 2 and 3 are almost identical (they differ by one word: "various"). Table 2 compares the results (number of clusters) for the 2 algorithms.

8. Table 3 raises questions for the reasons mentioned above. It is not clear how 4 out of 5 indices were applied.

9. Table 4 should explain the derived numbers.

10. In Table 6, there are typos in the names of the applied clustering algorithms: "Reference [25]+" and "Reference [27]+". Apparently, the number needs to be increased by 2.

11. The design of figure 1 is confusing.

12. From the diagram in Figure 2, it is not clear how the optimal number of clusters is determined from the results of comparing different CVIs. Perhaps the authors incorrectly presented the description in the blocks.

13. The quality of all drawings needs to be improved. They are fuzzy. Maybe some color needs to be added to the graphs.

The level of English is acceptable.

The main question that we need to try to answer is why these indices were chosen for comparison? How are indexes related to fuzzy clustering used if the improved clustering algorithm itself is crisp?

Experimental design

It is problematic to evaluate the results of experiments, because it is not clear how the fuzzy clustering quality indexes were used to evaluate the quality of crisp clustering.
Perhaps the authors could explain this.

Validity of the findings

No comments

Cite this review as

Reviewer 3 ·

Basic reporting

1. I suggest the authors include in the related work section.
2. The research question is not well delineated in the manuscript.

Experimental design

1. I suggest the authors explain the parameters values used in AP algorithms. How is the presented approach affected by other parameter values? How are they chosen?
2. I suggest the authors present a more illustrative method.
3. Should be nice to see the performance of the proposed method with challenge scenarios.
4. I really appreciate the description of the routines and their steps. Consequently, we must give the reader every opportunity to evaluate the relative strengths and weaknesses of various approaches depending on their research objective (s). Therefore, I would like to suggest that the authors add different ways of how improving the flaws for a particular method that still need to be explored. This could be added in the manuscript as future work and/or recommendations.

Validity of the findings

1. Response to conclusions is too obvious, please add discussion about them. In addition, I would encourage the authors to put the concluding remarks in a broader context: how do these minor points contribute to putting your work in a broader perspective?
2. Conclusions: Many affirmations included are not rigorous enough to address the goals and support the main conclusions. For example, “the new internal evaluation indices solve the possible problems with classical internal indices, such as the occurrence of invalid clustering results due to the excessive monotony of evaluation indices and the use of an excessively large number of 520 clusters. The experimental results show that the CEMS has high practicability and universality, and the CVI can evaluate the clustering quality and give an optimal clustering space more accurately when combined with other clustering algorithms”. I suppose that is necessary too much experiments to give a conclusion as that.

Additional comments

no comments

Cite this review as

---

## Round 0.2 · Minor Revisions

Dear Professor:
I have received the review reports for your paper submitted to PeeJ from the reviewers. According to the reports, I will recommend minor revision to your paper. Please refer to the reviewers’ opinions to improve your paper. Please also write a revision note such that the reviewers can easily check whether their comments are fully addressed. We look forward to receiving your revised manuscript soon.

Reviewer 1 ·

Basic reporting

1. To make it easier for the reader to read, position the chart near the reference and attach the corresponding footnote. Place the two tables on one page.
2. f(x,vi,σi) in the numerator and denominator of formula 20 is not defined and should be num(x,vi,σi).
3, the title of the table and the table are in two pages respectively, should be adjusted.
4. The horizontal and vertical headings in Figure3 should be identified.
5. The layout of the literature should be adjusted, for example, the position of the citations 17, 22, 35 and 36 is different from other citations.
6. the corresponding code should be given in the paper.

Experimental design

no comment

Validity of the findings

no comment

Additional comments

no comment

Cite this review as

Reviewer 3 ·

Basic reporting

The review is ok

Experimental design

The experiments were well prepared

Validity of the findings

The findings were well validated

Additional comments

All review is well done.

Cite this review as

---

## Round 0.3 · accepted · Accept

The authors have addressed the comments of the reviewers. I would suggest accepting the paper.